# Ultrasound during the COVID-19 Pandemic: A Global Approach

**DOI:** 10.3390/jcm12031057

**Published:** 2023-01-29

**Authors:** Anna Lombardi, Mariarosaria De Luca, Dario Fabiani, Francesco Sabatella, Carmen Del Giudice, Adriano Caputo, Luigi Cante, Michele Gambardella, Stefano Palermi, Rita Tavarozzi, Vincenzo Russo, Antonello D’Andrea

**Affiliations:** 1Department of General Medicine, San Leonardo Hospital, 80053 Castellammare di Stabia, Italy; 2Department of Translational Medical Sciences, University of Naples Federico II, 80131 Naples, Italy; 3Department of Cardiology, Luigi Vanvitelli University–Monaldi Hospital, 80131 Naples, Italy; 4Department of Advanced Biomedical Sciences, University of Naples Federico II, 80131 Naples, Italy; 5Public Health Department, University of Naples Federico II, 80131 Naples, Italy; 6Department of Translational Medicine, Università degli Studi del Piemonte Orientale, 28100 Novara, Italy; 7Department of Cardiology, Umberto I Hospital, 84014 Nocera Inferiore, Italy

**Keywords:** COVID-19, ultrasound, echocardiography, lung ultrasound, intensive care unit

## Abstract

SARS-CoV-2 (severe acute respiratory syndrome Coronavirus-2) rapidly spread worldwide as COVID-19 (Coronavirus disease 2019), causing a costly and deadly pandemic. Different pulmonary manifestations represent this syndrome’s most common clinical manifestations, together with the cardiovascular complications frequently observed in these patients. Ultrasound (US) evaluations of the lungs, heart, and lower limbs may be helpful in the diagnosis, follow-up, and prognosis of patients with COVID-19. Moreover, POCUS (point-of-care ultrasound) protocols are particularly useful for patients admitted to intensive care units. The present review aimed to highlight the clinical conditions during the SARS-CoV-2 pandemic in which the US represents a crucial diagnostic tool.

## 1. Introduction

SARS-CoV-2 (severe acute respiratory syndrome Coronavirus-2) rapidly spread across the globe starting in December 2019 as COVID-19 (Coronavirus disease 2019), causing deaths, huge health costs, and great variability in clinical manifestations, making its pathogenicity even more difficult to understand.

COVID-19 is a systemic infection, with multiple extrapulmonary manifestations beyond SARS: cardiovascular, hematological, renal, gastrointestinal, hepatobiliary, endocrinologic, dermatologic, neurologic, and ophthalmologic [1].

From an asymptomatic state to acute respiratory distress syndrome and multi-organ dysfunction [2], pulmonary manifestations represent the most impactful clinical findings, together with the cardiovascular impact that this virus continues to have. Ultrasound examination (US) became a cornerstone in the clinical evaluation of these patients, used at the bedside or in patients’ homes, with portable instruments, enabling the identification, differentiation, and follow-up of associated SARS-CoV-2 pathologies [3].

Therefore, the present review aimed to highlight the clinical conditions during the SARS-CoV-2 pandemic in which the US represents a valid diagnostic tool.

## 2. Lung Ultrasound

Pulmonary involvement during the COVID-19 pandemic is one of the main aspects studied due to its frequency and impact on severity, use of resources, and mortality of patients. However, respiratory manifestations are different, and are classified as mild, moderate, and severe, with possible progression [4].

While mild forms are mainly represented by upper respiratory tract symptoms including sore throat, mild cough, and loss of smell and taste, moderate forms usually manifest as bilateral interstitial pneumonia—frequently as a bilateral ground-glass appearance on a computed tomography scan—with the possibility of presenting with hypoxemia without dyspnea or signs of respiratory distress. In severe forms, pulmonary involvement usually evolves into acute respiratory distress syndrome (ARDS) with the need for admission into the intensive care unit (ICU) and mechanical ventilation support; organ failure also occurs, more frequently in elderly and obese people. ARDS, severe COVID-19 pneumonia, and multiple organ failure represent the most common causes of mortality in COVID-19 [4,5,6,7,8].

The entry gate of the virus into the cells is represented by the ACE2 receptors located in type II pneumocytes, involved in producing the pulmonary surfactant, subsequently causing ARDS through destruction and damage of alveolar cells, associated with decreased pulmonary surfactant and the cytokine storm, a hyperimmune response that leads to an excessive release of proinflammatory cytokines with consequent multiple organ failure [4,9,10].

Lung ultrasound (LUS) has proved to be useful in recent decades for detecting and monitoring pulmonary manifestations, with a renewed interest and consecutive deepening of knowledge in the COVID-19 era. US can potentially be useful, especially in resource-limited settings and moderate and severe forms of respiratory disease. Even if the lung parenchyma is not directly explorable by the US, due to the air itself and the limits of the thoracic cage, the artifacts generated by pathological pulmonary involvement provide important acoustic information [3].

The probe is positioned through the intercostal spaces. The choice of the transducer must be individualized based on the specific clinical case, considering that the linear probe uses high frequencies (7–10 MHz) which are optimal for the study of surface structures (pleural and subpleural), while the convex probe uses lower frequencies (3.5–5 MHz) and is indicated for the study of deep structures (diaphragm) and is also preferred in patients with obesity or a thick chest wall. It should be noted that LUS can detect only peripheral inflammatory lesions or those that extend onto the surface of the lung [3,11]. Approach possibilities include completing the chest examination to obtain complete ultrasound objectivity, examination directed by symptoms or regional auscultatory or percussor findings, then a selective examination of “hot zones” most likely to be sites of anomalies [3,12].

SARS-CoV-2 usually results in bilateral interstitial pneumonia, involving mainly the peripheral areas of the lung. Typical US signs are multiple confluent B lines (white lung), bilateral and more frequent in the posterior-basal site, generally asymmetrically distributed without a gravimetric gradient of distribution and with the presence of saving areas, peripheral parenchymal consolidations with variable dimensions and distribution reflecting the interstitial involvement, irregular pleural lines with thickening and discontinuity, reduction, or abolished sliding—especially in cases of more extensive parenchymal involvement—or with subpleural associated consolidations of small size and variable shape, and bilateral pleural effusion may also be associated [3,13,14].

Different methods have been proposed to quantify the involvement of pulmonary disease, including dividing the hemithorax into single areas in which to quantify the inflammatory involvement. Several studies indicate that baseline LUS score correlates with the eventual need for ICU admission and invasive mechanical ventilation, and the score is a predictor of mortality [11,15,16]. Even before that, LUS protocols and algorithms had been used for triage at admission, allowing diagnosis of COVID-19 pneumonia in patients with normal vital signs, although without standardization [17].

Finally, sequential lung point-of-care ultrasound (POCUS) performed with portable handheld devices represents a valid tool for monitoring patients in home isolation. Although data remain limited, the technique appears useful for early identification of pulmonary involvement, with results showing that B lines are present in early illness, but that resolution of sonographic findings is delayed in respect to symptom improvement [18].

## 3. Echocardiography

Several cardiac manifestations have been reported in COVID-19 cases: arrhythmia, hypertension, palpitations, myocarditis, myocardial injuries, cardiomyopathies, heart failure, and sudden cardiac death. Due to the presence of ACE2 receptors in the cardiac tissue and the cytokine storm, some of the medications that have been used to treat COVID-19 could induce arrhythmias [4,19,20].

Hypoxia has been considered a cause of cardiac arrhythmias and type II myocardial infarction (MI) in COVID-19 patients. MI in COVID-19 patients may present as type I, resulting in the rupture of plaque thrombus leading to occlusion of the coronary artery, or it may be the result of severe hypoxia leading to an ischemic type II MI. Cardiovascular manifestation of COVID-19 may present initial symptoms such as palpations, new diagnoses of hypertension, or chest pain, and may also present as late symptoms or complications [4,21].

Several POCUS protocols have been developed in different fields of cardiology [22].

According to the European Society of Cardiology (ESC) Task Force document [23] and the American Society of Echocardiography (ASE) [24], in COVID-19 patients with cardiac involvement, transthoracic echocardiography (TTE) should be based on POCUS both to answer specific clinical questions and to minimize patient contact with machines and operators [24], and a comprehensive TTE exam should integrate the initial evaluation to better characterize COVID-19-related cardiac involvement [25].

The ESC Working Group on Myocardial and Pericardial Diseases recommended TTE as part of the initial diagnostic workup for all COVID-19 patients with suspected myocarditis [26,27], although the potential role of the electrocardiography in these patients should not be underestimated [28]. Two-dimensional (2D) echocardiographic findings in COVID-19 myocarditis include increased endocavitary dimensions in the left (LV) or right ventricle (RV), normal or increased wall thickness, global or regional ventricular systolic dysfunction, diastolic dysfunction, LV thrombus, and pericardial involvement with effusion [29]. These features are not particular to COVID-19 but are found in common in viral myocarditis due to other etiological agents. Other TTE techniques including tissue Doppler imaging (TDI) and contrast echocardiography (CE) could be used to further the stratification of COVID-19 myocarditis prognosis [30]. TDI is usually impaired in patients with acute myocarditis [31], while CE as a diagnostic tool allows better evaluation of regional LV systolic function and identification of LV thrombi [32].

Moreover speckle-tracking echocardiography (STE), an ultrasound tool that provides a quantitative, objective, non-doppler assessment of systolic and diastolic function by evaluating the movement of speckles on 2D echocardiographic images, has shown higher diagnostic accuracy than standard TTE in the examination of early systolic dysfunction in the setting of COVID-19 myocarditis [33,34,35]. Furthermore, STE reduction in COVID-19 myocarditis has a typical deformation pattern, featuring the involvement of the sub-epicardial layer rather than a sub-endocardial portion of the myocardial wall [36]. The tendency to spare sub-endocardial regions suggests the exclusion of ischemic damage and allows the possibility of a differential diagnosis, particularly in the subset of myocarditis with clinical presentation of acute coronary syndrome [36]. Although according to ESC guidance [37], CMR represents the main diagnostic test for COVID-19 myocarditis detection, recently STE has acquired an emerging role due to its capacity to identify subclinical COVID-19 myocardial injury in an intensive care setting, where a bedside assessment is more easily accessible [38].

Beyond cardiac conditions directly caused by the COVID-19 virus, the pandemic has also caused an increase in MI-related complications and fatal cardiac events [39]. This is partly because initial symptoms were often misinterpreted by the patients themselves, and secondly because people avoided contact with health professionals and emergency departments for fear of contagion. Overall, the time from symptom onset to first medical contact resulted in delayed diagnosis and worse outcomes [39].

In fact, during the COVID-19 pandemic, in addition to the reduced number of hospitalizations for MI [39,40], Bryndza et al. [41] observed a corresponding decrease in the numbers of PCI procedures performed in patients with NSTEMI and STEMI [42], a significant increase experience of pain longer than 12 h, and up to twice as many mechanical complications and MI related major complications [43].

## 4. Vascular Ultrasound

The endothelium is a target for the COVID-19 virus, both directly because the virus infects endothelial cells, and indirectly through the prothrombotic status of the acute phase response. Endothelial ACE2 receptor binding starts a complex inflammatory response with endothelial cell activation, characterized by high levels of the platelet adhesion molecule von Willebrand factor and low levels of disintegrins and metalloproteinase with thrombospondin motifs 13 (ADAMTS13), increased expression of tissue factor, decreased expression of tissue factor pathway inhibitor, and an impaired protein C pathway. This is the base for the development of a pro-coagulant state known as COVID-19-associated coagulopathy [44,45,46,47,48,49].

The hypercoagulable condition, endothelium activation, and immobility of the patient increase the risk of thromboembolic complications, including microvascular thrombosis, venous and pulmonary thromboembolism, and (to a lesser extent) acute arterial thrombosis, especially stroke [44]. A recent meta-analysis evaluating the incidence of venous thromboembolism among hospitalized patients with COVID-19 showed that the overall incidence rate was 17.3%, with about two-thirds of the events being deep vein thrombosis [50]. Pharmacological and mechanical thromboprophylaxis are used as preventive strategies according to recommended practice, and early mobilization is promoted whenever possible [51]. The presence of lower-limb thrombus should be evaluated by vein compression, color Doppler imaging, and spectral Doppler waveforms, from the inguinal ligament through the popliteal vein to the calf vein confluence, as recommended [52]. In this scenario, studies have assessed the usefulness and feasibility of duplex ultrasonography screening protocol and the characteristics of deep vein thrombosis [53]. The routine use of compression ultrasound has been recognized as a useful tool to investigate the presence of deep venous thrombosis in SARS-CoV-2 critical and non-critical patients [54,55]. The most involved district appears to be the lower limb, particularly the left, while the upper limbs are less frequently involved. On both sides, the distal section of the popliteal vein was the most common site of thrombosis [56]. Voicu S. et al. [57] conducted a prospective cohort study aimed at evaluating rheological conditions in mechanically ventilated COVID-19 patients, recognizing the larger diameter of the common femoral vein and lower peak blood flow velocities as potential predisposing local factors of thrombotic complications. However, bilateral ultrasonography of lower limb screening protocol is not useful in stable COVID-19 patient populations without symptoms of vein thrombosis [58].

Data on arterial thrombosis in COVID-19 patients are scarce. According to a recent systematic review [45], arterial thrombosis seems to occur approximately in 4% of critically ill COVID-19 patients (especially men, the elderly, and patients with comorbidities). It was symptomatic in >95% of cases, with multiple arterial involvements in approximately 18% of patients, mostly in limb arteries (39%), cerebral arteries (24%), great vessels (aorta, common iliac, common carotid, and brachiocephalic trunk; 19%), coronary arteries (9%), and the superior mesenteric artery (8%). The color Doppler ultrasound can demonstrate occlusion through a pulsed Doppler waveform, showing the absence of velocimetric curve at the level of the occlusion: when the occlusion is recent, the arterial lumen appears occupied by hypoechoic material, becoming hyperechoic when the occlusion is long-standing, with associated alterations in the thickened vessel wall and reduced lumen [59]. Transcranial Doppler ultrasound (TCD) is a portable and noninvasive technique that can detect circulating emboli and search for cardiac embolism or vascular stenosis as potential stroke mechanisms [60,61]. Ischemic stroke, especially from large vessel occlusion, represents a potential neurological complication of COVID-19 infection [60,62,63]. In COVID-19-positive stroke patients, relatively low cerebral blood flow velocities were observed by TCD, despite low hematocrit and good left ventricular ejection fraction, correlated with arterial oxygen content and C-reactive protein [60].

TCD has also been employed to study the changes in brain vasoreactivity after COVID-19 infection, and thus to define chronic endothelial dysfunction. In a cross-sectional study, Marcic et al. [64] used transcranial color-coded Doppler ultrasound together with a breath-holding test (BHT) to examine cerebral vasoreactivity and brain endothelial function, analyzing the parameters of the flow rate through the middle cerebral artery (MCA) to calculate the breath-holding index (BHI), a method to assess cerebrovascular reactivity. Subjects had lower measured velocity parameters through MCA at rest period and after BHT, lower relative increases of flow velocities after BHT, and lower BHI. They still had impaired cerebral vasoreactivity 300 days after COVID-19 infection, thus developing chronic endothelial dysfunction.

The main pulmonary and cardiovascular ultrasound findings associated with COVID-19 are collected in Table 1. Pathological cardiac, thoracic, and vascular effects correlated with ultrasound changes caused by COVID-19 are summarized in Figure 1.

## 5. Ultrasound in ICU

Imaging aimed at patients admitted to the ICU must respond to some practical needs: it must employ easily transportable instruments, preferably applicable to unstable and non-transportable patients, easily disinfectable, capable of rapid application with the minimum number of healthcare professionals involved in the examination to reduce the exposure risk. POCUS has all these characteristics and this is why it has been widely used during the COVID-19 pandemic [65]. Pocket-carried wireless ultrasound devices should be preferentially used, as they can be protected with a probe cover and easily cleaned. Portable machines should be exclusively dedicated to COVID-19 patients and employed always with the maximum care for sterilization, preferable used with touch-screens and single-use gel packets [24,66,67,68,69].

### 5.1. Lung Ultrasound

Lung POCUS in ICU-ventilated patients can be useful to confirm endotracheal intubation. A linear ultrasound probe should be placed transversely on the anterior aspect of the neck at the level of the cricothyroid membrane. Through accurate visualization of the pleural movements during the ventilatory cycle, US moreover allows the identification of selective intubation [70,71]. Even more specific is the US study of the diaphragm, which holds the possibility of monitoring patients while enabling their liberation from mechanical ventilation [65,72]. The M-mode study of diaphragmatic kinetics uses US to represent the beginning and end of a patient’s respiratory effort: it is useful for recognizing diaphragmatic dysfunction, and pulmonary, cardiac, and hemodynamic signals of ineffective weaning from ventilation treatment, with predictive ability for post-extubation respiratory distress [72]. In addition, the US evidence of diaphragm-thickening fraction, together with the parasternal-thickening fraction, can predict weaning failure or the spontaneous breathing trial [17,73].

US also allows the detection of complications associated with ventilation, such as pneumothorax. When typical US findings of pneumotorax are detected (absence of pleural sliding, B lines, and lung pulse, together with the presence of a barcode sign and pathognomonic lung points) [3,74,75], the LUS shows a negative predictive value for pneumothorax of approximately 100% [17,76]. Furthermore, the appearance of large consolidations in patients in mechanical ventilation, not responsive to recruitment maneuvers or suction, are highly suggestive of secondary bacterial infection such as ventilator-associated pneumonia [17,77,78,79].

### 5.2. Cardiovascular Ultrasound

Some of the main echocardiographic features of COVID-19 include but are not limited to (I) hyperdynamic cardiac function, (II) acute stress-induced cardiomyopathy, (III) right ventricular enlargement, and acute pulmonary hypertension [67] (Figure 2). Therefore, cardiac POCUS can be useful to monitor the cardiovascular and hemodynamic status of ICU patients [65,67,80], paying attention to the particular conditions as previously extensively described in the dedicated paragraph. Although transthoracic echocardiography can provide the information needed in most patients, transesophageal echocardiography can be useful to evaluate hemodynamic instability during prone ventilation, to perform serial evaluations of the lungs, to support cardiac arrest resuscitation, and to guide veno-venous ECMO cannulation [81].

Thrombotic complications such as pulmonary embolism (PE) are common in COVID-19 patients, especially in ICU [82]. POCUS ultrasound is a highly recommended tool for orientation in critically ill patients with suspected or confirmed complications. Some echocardiographic findings were found to be strictly related to a diagnosis of PE, such as RV size [83] and disfunction [84]. Polito et al. [85], for example, found that RV systolic dysfunction, increased systolic pulmonary artery pressure (PAPS), and poor RV-arterial coupling may help to identify COVID-19 patients at higher risk of mortality and PE during hospitalization.

Therefore, POCUS has also been proposed to guide vascular access in COVID-19 patients, representing a tool for fast screening of circulatory and fluid repletion status, identifying types of shock, and thrombosis screening [65,67,86,87], considering that venous thromboembolic events occur in about 28% of ICU hospitalized patients [44,50].

### 5.3. Ultrasound in Shock and Fluid Administration

A combined ultrasound evaluation of the inferior vena cava (IVC), heart, and lungs allows etiological classification of the type of the shock [3].

Increased IVC diameter, congestive status of the lungs (B lines), and the hyperpulsatility of portal vessels and hepatic veins suggest excessive fluid filling or high right atrial pressure, a possible cause of obstructive shock (i.e., pulmonary embolism) [3,87]. Conversely, a dilated IVC with pulmonary A-lines and lung points describes an obstructive shock caused by pneumothorax [3].

When the IVC appears collapsed, it is probably a case of hypovolemic shock, so pulmonary B lines are not expected, but rather cardiac wall “kiss” effect is present; moreover, we could visualize lung consolidations as a sign of pneumonia, with bacterial over-infection that could lead to septic shock [3].

Fluid management is fundamental in critically ill patients: fluid overload can exacerbate lung dysfunction in COVID-19 patients. Nevertheless, the interpretation of inferior and superior vena cava collapsibility is difficult during a ventilation modality [17,88,89,90,91]. In these cases, TTE can assess other dynamic index predictors of fluid responsiveness, based on variation in stroke volume, passive leg raising, and mini-bolus administration techniques [17,92,93]. Similarly, LUS can provide information on fluid tolerance and detect excessive fluid resuscitation [17,93,94]. Nevertheless, B lines are present in both interstitial pneumonia and pulmonary edema, and so it is important to differentiate the two conditions. In lung-fluid overload, B lines show a gradient distribution: they are bilateral and without areas of sparing, with a regular pleural line and usually pleural effusion. Differently, LUS characteristics of interstitial pneumonia, previously described, include pleural thickening and irregularity, and the irregular distribution of B lines and saving areas [3].

### 5.4. POCUS Protocols and Other Applications

Several integrated POCUS protocols have been proposed to allow global assessment of COVID-19 patients, especially in critical conditions [65], from a bedside tri-POCUS approach that assesses lungs, heart, and venous system in evaluating the volume status [95] to a complete sequential method, easily remembered as COVID protocol [86] which provides several elements to evaluate each field to be studied (C–cardiac evaluation, O–output, V–ventilation, I–intubation, D–Doppler and Deep venous thromboembolism/pulmonary embolism).

In COVID-19 infection, POCUS has shown excellent application, allowing global framing of patients, quickly and at the bedside.

POCUS has also been proposed as a screening tool for high-risk dialytic patients, to allow for early recognition and management of COVID-19 [65,96]. Similarly, US has been applied to demonstrate resolution of B lines during hemodialysis [17,97]. Other potentially applications of POCUS in COVID-19 patients are related to secondary organ dysfunction diagnosis [17].

The US study of the optic nerve sheath diameter (ONSD) in COVID-19 patients admitted in ICU, for example, showed that an increased diameter caused by hypoxia and hypercapnia due to acute pulmonary failure and associated with higher intracranial pressure (ICP) may be related to poor prognosis and mortality [98]. Thus, even if elevated ONSD is common in severe COVID-19 infection and is associated with adverse outcomes, elevated ICP has not been extensively studied in severe COVID-19 infection, and ventilation modalities with the application of positive end-expiratory pressure can affect ICP and ONSD values [99,100,101]. 

Furthermore, the incidence of gastrointestinal involvement has ranged from 12% to 61% in patients with COVID-19, with anorexia, nausea and/or vomiting, diarrhea, and abdominal pain as the main manifestations [1]. Gastrointestinal involvement is the most common extrapulmonary manifestation, always due to the presence of ACE2 receptors [102,103]. While US has had a secondary role, the main radiological findings have been inflammation bowel signs: bowel wall thickening, submucosal edema, bowel dilation, pericolic fluid, and locoregional inflammatory lymph nodes. Mesenteric and portal vein thrombosis with secondary small bowel wall inflammation and ischemia, have also been typical imaging findings described in COVID-19 [102,104,105].

Other non-specific US applications concerning the extrapulmonary manifestations of COVID-19 infection may concern the inflammatory involvement of the pancreas and gallbladder, and acute renal failure, but the amount of data is negligible, and few studies have been conducted [102].

Comprehensive POCUS protocols suggested during the COVID-19 pandemic are summarized in Table 2.

## 6. Vaccination-Associated Reactive Lymphadenopathy

The introduction of vaccines marked the turning point in the fight against the COVID-19 pandemic, even if with some initial difficulties mainly related to some serious and sometimes fatal adverse reactions, such as the rare cases of vaccine-induced immune thrombotic thrombocytopenia and cerebral venous sinus thrombosis after viral vector vaccines. However, the most commonly reported adverse effects after COVID-19 vaccination consist of local reactions at the injection site followed by several non-specific flu-like symptoms [105,106,107,108,109,110].

Vaccination-associated reactive lymphadenopathy is included in local adverse reactions to vaccination. It is more commonly observed after receiving COVID-19 messenger RNA (mRNA) vaccines, because compared to protein-based vaccines, mRNA vaccines determine a more robust and rapid B-cell proliferation in the germinal centers of lymph nodes [111,112,113]. Moreover, younger age, first dose and an interval of 4 weeks represent important factors in developing axillary lymphadenopathy [114]. Axillary swelling or tenderness has been reported in up to 16% of recipients of the Moderna vaccine [114,115], with imaging findings of axillary lymphadenopathy in up to 15% of recipients of the Pfizer-BioNTech vaccine and 57% of Moderna recipients [111,116]. Sixty-one percent of axillary lymphadenopathies were detected with the US alone [78,84], even if most patients demonstrate subclinical lymphadenopathy on imaging [111,114].

US is a rapid, easily available, and user-friendly method to evaluate lymph node enlargement, structure, and vascularization [117,118]. Typical abnormalities found in vaccination-associated axillary lymphadenopathy are lymph node enlargement with cortical thickening >3 mm, with or without a preserved fatty hilum on US examination [111]. However, the loss of the central fatty hilum, together with a rounded hypoechoic aspect, increased cortical flow, and increase in the overall number of lymph nodes, represent suspicious morphologic characteristics [111,114,118]. Moreover, in a study of US scans of temporal changes in imaging characteristics, only 26% of women reached the complete resolution of axillary lymphadenopathy at a median of 6 weeks after vaccination, and persistent lymphadenopathy was observed in 51% after 16 weeks [111]. Follow-up US examination at least 12 weeks after vaccination may be reasonable, considering vaccine type and time elapsed since vaccination [111,114].

## 7. Conclusions

Ultrasonography is a suitable tool for assessing the effects of COVID-19, a virus with multi-organ tropism that can cause direct and indirect damage to several tissues and organs. Ultrasound evaluations are essential in patients where restrictions on transport and isolation procedures hinder the diagnostic process. Ultrasonographic analysis of the lungs, heart, and lower limbs of COVID-19 patients can clarify the prognosis of these patients and support the establishment and monitoring of therapies. However, the protocols to establish reproducibility, scoring systems, and predictive value in COVID-19 patients are not yet well defined, and further studies will be essential to standardize procedures.

## Figures and Tables

**Figure 1 jcm-12-01057-f001:**
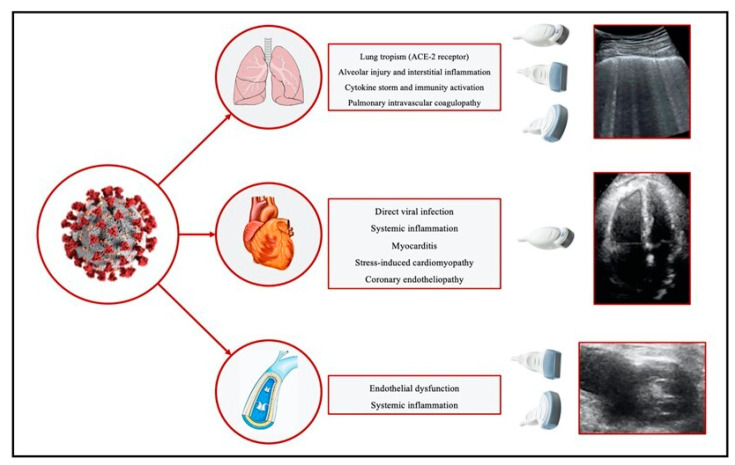
Main pathological cardiac, thoracic, and vascular effects caused by COVID-19 correlated with ultrasound changes. ACE: Angiotensin-converting enzyme.

**Figure 2 jcm-12-01057-f002:**
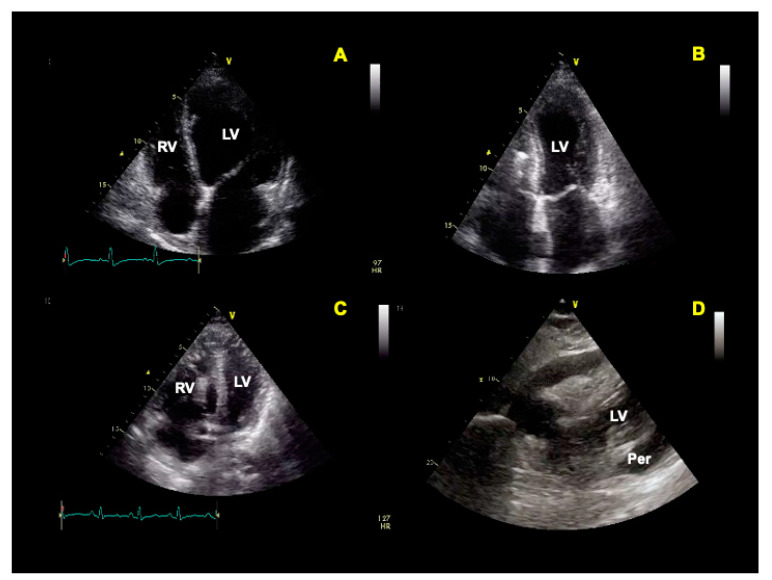
Possible features of echocardiography in COVID patients. (**A**) Left ventricle (LV) dilatation and dysfunction; (**B**) Takotsubo cardiomyopathy; (**C**) right ventricle (RV) dilatation secondary to pulmonary embolism; (**D**) pericardial (Per) effusion.

**Table 1 jcm-12-01057-t001:** Common pulmonary and cardiovascular ultrasound findings in COVID-19.

Lung	Heart	Veins
Bilateral multiple or confluent B lines.	Left or right ventricle increased endocavitary dimensions	Thrombosis ultrasound findings: non-compressible venous segment; loss of phasic flow on Valsalva maneuver; absent color flow; lack of flow augmentation with calf squeeze; increased flow in superficial veins
Peripheral parenchymal consolidations with distribution reflect interstitial involvement.	Normal or increased wall thickness	Larger diameter of common femoral vein and lower peak blood-flow velocities
Irregular pleural line with thickenings and discontinuity	Global or regional ventricular systolic dysfunction	
Bilateral pleural effusion	Diastolic dysfunction	
	Left ventricle thrombus	
	Pericardial involvement with effusion	

**Table 2 jcm-12-01057-t002:** COVID-19 comprehensive POCUS protocols.

First Author, Year of Publication (Ref.)	Article Type	Setting	POCUS Protocols
Koratala A, 2020 [90]	Editorial	Critically ill Patients with COVID-19	Tri-POCUS approach: Lungs → B-line numbers and distribution, pleural line abnormalities, consolidations, pleural effusion Heart → pericardial effusion, LV EF, RV relative size, passive leg raise and stroke volume assessment, inferior vena cava collapsibility Venous system
Anile A, 2020 [81]	Commentary	Critically ill Patients with COVID-19	COVID-US approach: C: cardiac evaluation O: outputs → renal resistive index, velocity-time integral V: ventilation → B-line patterns, hyperinflation and recruitment response, lung score, pneumothorax/effusion. I: intubation → prediction of difficult laryngoscopy/intubation, endotracheal intubation confirmation D: Doppler and deep venous thromboembolism/pulmonary embolism
Coneybeare, 2021 [101]	Review	Patients under investigation for COVID-19	COVUS approach: Cardiac US → parasternal long, parasternal short, apical 4-chamber, subxiphoid, and inferior vena cava Lung US → Bilateral anterior, bilateral axillary, and bilateral posterior (lawnmower technique for assessment)
Toraskar K, 2022 [102]	Cross-sectional study	Critically ill Patients with COVID-19	Lung US → BLUE protocol [103] Cardiac US → FATE protocol [104] Lower limb deep vein US: two-point compression sonography method

US: ultrasound; LV: left ventricle; RV: right ventricle, EF: ejection fraction.

## Data Availability

No new data were created for the aim of the present article.

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
