# Peer review of "Ultrasound during the COVID-19 Pandemic: A Global Approach"

_jcm, 2023, doi:10.3390/jcm12031057_

Round 1
Reviewer 1 Report
Abstract: “Ultrasound (US) evaluations of the lungs, heart and lower limbs are essential in diagnosis, follow-up and prognosis of patients with COVID-19.” This statement isn’t really supported by the weight of the evidence presented in the paper. Broadly speaking, the use of diagnostic ultrasound in COVID-19 is still in its infancy and evidence of benefit is still largely lacking, especially with regard to the diagnostic value-add of lung ultrasound over CXR/CT in COVID-19 patients. To acknowledge this, I would change this sentence to something like: “Ultrasound (US) evaluations of the lungs, heart and lower limbs may be helpful in the diagnosis, follow-up and prognosis of patients with COVID-19.”
Line 56: the terms “silent pneumonia” and “silent hypoxia” aren’t widely used where I practice. Could the authors clarify what they mean here?
Lines 69-70: This hasn’t really been my clinical experience or the experience of my other ICU colleagues. In severe COVID-19 patients requiring ICU-level care, the ultrasound can serve as an optional adjunct to other imaging options such as CXR and CT scans, but has rarely been “critical” in my care of these patients, even though I perform diagnostic ultrasound on >= 25% of my ICU patients in general, routinely use lung ultrasound specifically, and am a nationally recognized expert in ICU-relevant diagnostic POCUS. Further, as the authors themselves acknowledge in the conclusion of the paper, the exact role of diagnostic POCUS in COVID managements remains uncertain. Consequently, I would change the wording from “critical” to “potentially useful, especially in resource-limited settings.”
Lines 80-82: The authors are defining the term “point-of-care ultrasonography” in a way that deviates from existing published peer-reviewed literature. In the existing literature (e.g., PMID: 35582953; PMID: 34670045), a point-of-care ultrasound exam is one that is performed and interpreted by a patient’s bedside treating provider. This is in contrast to consultative ultrasound, which is requested by a bedside treating provider but performed by a separate specialist team. So a diagnostic lung POCUS and a consultative lung ultrasound can EACH be either limited or comprehensive in scope. An exam being limited in scope is NOT what defines it as a point-of-care exam per the nomenclature already in use in the peer-reviewed literature.
Author Response
Please find attached the response to your reviews, thanks

Reviewer 2 Report
Lombardi et al. discussed the use of ultrasounds during the COVID-19 pandemic. The ultrasound evaluations of the lungs, heart, and lower limbs were essential in the diagnosis, follow-up, and prognosis of patients with COVID-19. Of interest, the authors aimed to highlight the clinical conditions in which the US represents a crucial diagnostic tool during the SARS-COV-2 pandemic and when their use could be used to stratify the prognosis of these patients and establish the therapies.
Also, ultrasounds were pivotal in the diagnostic work-up of patients with COVID-19, where restrictions on transport and isolation procedures hinder the diagnostic process. The review is of interest nevertheless, I have some concerns that need to be addressed before the study could be re-submitted.
1. Can the authors specify in which cases the trans-thoracic echocardiogram during COVID-19 may raise the suspicion of pulmonary embolism and how to act in this case?
2. Can the authors comment on the role of the TAPSE/sysPAP ratio in patients with COVID-19?
3. Can the authors discuss the utility of speckle tracking echocardiography in identifying subclinical myocardial dysfunction in young adults recovered from mild COVID-19 and integrate the following reference PMID: 35906710?
4. Can the authors comment on what the EKG-graphic changes are during COVID-19 that may require further diagnostic investigation with an echocardiogram? Please integrate with this point the following reference PMID: 33512742.
5. Any comments about TEE? In which cases, could it be mandatory, despite SARS-COV-2 infection?
6. I suggest improving the Figure Legend of Figure 2.
Author Response

(The authors gave the same response as above.)

Round 2
Reviewer 2 Report
Thanks for addressing all my comments.
Author Response
Thanks!